# Psychiatric Disorders and Cognitive Fluctuations in Parkinson’s Disease: Changing Approaches in the First Decades of the 21st Century

**DOI:** 10.3390/brainsci14121233

**Published:** 2024-12-08

**Authors:** Marco Onofrj, Matteo Alessandro De Rosa, Mirella Russo, Paola Ajdinaj, Dario Calisi, Astrid Thomas, Stefano Luca Sensi

**Affiliations:** 1Department of Neuroscience, Imaging and Clinical Sciences, “G. d’Annunzio” University of Chieti-Pescara, 66100 Chieti, Italy; matteo.derosa1994@libero.it (M.A.D.R.); mirella.russo.92@gmail.com (M.R.); paola.ajdinaj@gmail.com (P.A.); dariocalisi95@outlook.it (D.C.); athomas@unich.it (A.T.); ssensi@uci.edu (S.L.S.); 2Center for Advanced Studies and Technology (CAST), “G. d’Annunzio” University of Chieti-Pescara, 66100 Chieti, Italy; 3Neurology Institute, SS. Annunziata University Hospital, “G. d’Annunzio” University of Chieti-Pescara, 66100 Chieti, Italy; 4Institute for Advanced Biomedical Technologies (ITAB), “G. d’Annunzio” University of Chieti-Pescara, 66100 Chieti, Italy

**Keywords:** Parkinson’s Disease, functional neurological disorders, psychiatric symptoms, cognitive impairment

## Abstract

Parkinson’s Disease (PD) is a multifaceted neurodegenerative disorder characterized, in addition to the well-recognized motor disturbances, by a complex interplay between cognitive and psychiatric manifestations. We dissect the complex landscape of PD-related psychiatric symptoms, taking into account the impact of functional neurological disorders, somatic delusions, impulse control disorders, and conditions within the bipolar spectrum. The newer entities of somatoform and functional neurological disorders, as well as preexisting bipolar spectrum disorders, are analyzed in detail. Moreover, we emphasize the need for a holistic understanding of PD, wherein the cognitive and psychiatric dimensions are valued alongside motor symptoms. Such an approach aims to facilitate early detection and personalized interventions, and enhance the overall quality of life for individuals suffering from this neurodegenerative disorder.

## 1. Introduction

Parkinson’s Disease (PD) is one of the most prevalent neurodegenerative disorders; a rising trend in its prevalence has been reported by several studies [1,2,3], while other studies have challenged these conclusions, highlighting the effect of population growth and patterns of age distribution [4]. PD is characterized by well-known motor symptoms, as well as non-motor symptoms [5,6], which encompass neuropsychiatric and cognitive disturbances [7,8]. These symptoms significantly impact the patients’ therapeutical needs and their response upon the introduction of either standard treatments, such as dopamine-mimetic drugs, or advanced treatments, like parenteral gel therapies or deep brain stimulation [9,10,11,12,13]. Neuropsychiatric disturbances in PD are multifaceted and considerably more complex than reactive depression that occurs in response to the diagnosis [14,15,16]. They include disorders associated with an increased predisposition to develop PD, disorders that consistently accompany the course of PD, and disorders precipitated by therapies or occurring independently thereof.

Despite the fact that many earlier studies have focused on depression and cognitive assessments in PD [17], the identification of neuropsychiatric disorders in PD and parkinsonisms underwent substantial changes after the year 2000, as new neuropsychiatric disturbances were clearly identified, e.g., different kinds of hallucinations [18], transient oscillations of vigilance and cognitive functions [19], and somatic symptom and functional neurological disorders (SSD and FND, respectively) [20]. Also, impulse control disorders (ICDs) [21] were evidenced in early 2000, with a predominant approach linking their occurrence to treatment. Still, in the second decade of the century, a growing consensus supported the strong connection between PD and bipolar disorder [22]. This narrative review will focus on recent advancements in the understanding of neuropsychiatric symptoms associated with PD that have emerged in the first quarter of the century.

We use the term PD or “Typical Parkinsonism”. “Genetic Parkinsonism” is instead employed to indicate monogenic forms (e.g., Park1, Park2, etc.) or heterozygous mutations in Glucocerebrosidase (GBA) [23]. Finally, we employ the term “Atypical Parkinsonism” to indicate multiple system atrophy (MSA), progressive supranuclear palsy (PSP), and corticobasal degeneration (CBD). Additionally, we use the term “Atypical-Atypical Parkinsonism” to refer to forms that manifest as MSA, PSP, or CBS phenotypes, but also exhibit other symptoms, primarily cognitive or psychiatric. These forms are genetically determined by mutations in GBA, fragile X, cerebrotendinous xanthomatosis, and Niemann-Pick, as well as mitochondrial genetic mutations [24,25,26,27,28,29].

## 2. History

The assumption that “sensitivity and intelligence appear to remain intact” in PD, enclosed in quotation marks, originates from the original description by James Parkinson in 1817 [30] and has historically influenced clinical perceptions and treatment approaches. However, as early as the late 19th century, cases of PD with psychiatric or cognitive disorders were documented.

Following James Parkinson’s initial description and J.M. Charcot’s clinical studies [31], with the Spanish flu and the emergence of Post-Encephalitic Parkinsonism (PEP), mental disorders became increasingly recognized. At that time, the primary concern was the management of psychiatric disorders, including impulsive behavior, hypersexuality, and other compulsions like gambling, as well as delusions, hallucinations, and cognitive impairment [32]. Additionally, a high incidence of somatization disorders was observed, categorized at the time within the spectrum of “hysteria” [33]. Prominent neuropsychiatrists and neuropathologists in the years between 1920 and 1950 were particularly interested in patients with PEP who exhibited hysterical-type disorders, as they believed that PEP could serve as a model for identifying the organic basis of hysteria (experimental disease) [34,35,36,37,38,39]. In the same period, psychoanalysis was born, a construct that rapidly gained a dominant role in society. Consequently, the psychodynamic interpretation of hysteria became axiomatic, and the concept of experimental disease fell into oblivion [40,41]. We will discuss later how the hypotheses of that time have found answers in recent years through studies on different forms of PD.

Once the wave of patients with PEP subsided, the issue of psychiatric disorders in PD was largely overlooked as if it did not exist, only to resurface at the beginning of the 21st century. Nevertheless, in the late 1960s, when describing the effect of the then-new drug, L-Dopa, the writer J. Updike, in his tetralogy “Rabbit Redux”, detailed psychiatric disorders in a patient with PD, a sign of the presence of the problem [42].

In 2003, Y. Agid and his colleagues wrote “Parkinson’s Disease is a neuropsychiatric disorder” [43] because the evidence of hallucinations, delusions, behavioral disturbances, and executive function impairments had become undeniable. Common neurological knowledge shifted from the 1980s and early 1990s, when the appearance of psychiatric disorders in PD was considered a rare event worthy of diagnostic definition, to the early 2000s, when the London Brain Bank reported that if a patient diagnosed with idiopathic PD (Typical, as mentioned earlier) did not experience hallucinations during the disease, the diagnosis should be changed [44]. Agid’s study opened (or reopened) the discipline, and the neuropsychiatry of PD found enthusiastic adherents. A list detailing the principal neuropsychiatric disorders of PD and their associations with different stages of the disease is reported in Table 1.

## 3. The Pathophysiological Mechanisms Underpinning These Disorders

Numerous studies have attempted to elucidate the mechanism underlying the emergence of cognitive and psychiatric disturbances and the cerebral dysfunctions contributing to hallucinations and delusions [45,46,47,48,49,50,51,52,53]. In this section, we briefly outline their key characteristics, as numerous studies have recently converged to provide evidence of network dysfunctions that were lacking at the time of PEP.

Functional neuroimaging studies that highlight the disconnection between different brain networks in the context of PD and its associated psychiatric disorders provide valuable insights into the neural basis of these symptoms. The evidence suggests an alteration in communication among various brain areas involved in perception, attention processing, and the generation of internal narrative processes [45,46,50,51,54].

This disconnection seems to favor the prevalence of internal narrative processes over the monitoring of external reality, leading to the emergence of psychotic symptoms such as visual hallucinations, delusions, somatic disturbances, and overall disinhibition [18,49,51,55,56,57].

The origin of these alterations in functional connectivity is at the thalamic level, where alpha-synuclein deposits contribute to dysregulation of the thalamic pacing system, influencing cortical activity [46,50,56]. This dysregulation, initially episodic but progressively chronic, is associated with various cognitive and psychotic manifestations. An intriguing hypothesis is related to imbalanced phylogenetic evolution of the frontal lobes compared to subcortical structures [55]. This imbalance may render the frontal structures more vulnerable to neurodegenerative processes, thereby contributing to the observed symptomatology in patients with PD and psychiatric disorders. Furthermore, recent studies demonstrating similarities in network alterations across various psychiatric disorders suggest that these disorders may share underlying neural mechanisms, even though they manifest in different ways [58,59,60,61]. This supports the idea that different psychiatric manifestations could represent symptoms of a common dysfunction rather than distinct diseases.

Overall, these findings can contribute to a better understanding of the neural mechanisms underlying psychiatric symptoms in PD and underscore the need for innovative treatment modalities that address both cognitive and psychiatric complexities in PD.

## 4. Neuropsychiatric Disorders

### 4.1. Depression

Depression was the first neuropsychiatric disorder described in patients with PD [62,63]. Earlier studies, guided by the dominant psychoanalytic approach of their time, even hypothesized common etiologies [64,65,66]. Recent studies primarily utilize Depression Scales to evaluate PD patients [67]. PD patients typically experience only some of the symptoms of depression, such as anhedonia, hopelessness, psychomotor retardation (notably, apathy is no longer considered a feature of depression), and fatigue [68,69]. Still, they do not experience guilt, which is a core element analyzed in psychopathological studies [70,71]. In this review, depression in PD will not be further analyzed, as previous reviews have diffusely analyzed the overlap or comorbidity with PD [15,16,72].

### 4.2. Hallucinations and Delusions

Hallucinations in PD can occur in two typologies: the first, considered mild, may appear at the onset of the disease when motor symptoms are mild. Simple hallucinations can also happen in the absence of L-dopa/dopamine, typically consisting of illusions or passing hallucinations that lack a narrative and may disappear after a few months [18,73]. The second type is a complex hallucination, represented by animals or people, with a narrative (a story) and interaction between the patient and the hallucination. These appear several years after the onset of motor symptoms and are not dependent on L-dopa/dopamine pharmacokinetics [74]. However, they can be induced by the introduction of or increase in L-dopa in patients with PDD or DLB [16]. Initial insight may be preserved in this latter form, and if patients focus their attention on the hallucination, it may disappear. However, with the progression of the disease, insight is progressively lost, leading to the emergence of delusions and multisensory hallucinations. It should be emphasized that the gradual loss of insight follows the onset of cognitive deterioration. These two forms of hallucinations, simple and complex, rather than representing two separate entities, constitute more or less complex phenomena along a psychotic continuum that follows the disease’s progression [18,50]. The first type may also appear in Monogenic Parkinsonism and should not be present in Atypical Parkinsonism. The second type occurs in advanced Typical PD and Atypical-Atypical PD, and should not be present in Atypical Parkinsonism.

The prevalence of hallucinations throughout the disease varies between 80% and 100% [16,75,76].

Delusions are much less frequent and likely depend on similar mechanisms, with predisposition having a more significant impact. Genetic predisposition and the presence of psychiatric comorbidities explain their early onset [16,77]. Typically, delusions in PD primarily appear in advanced stages when cognitive decline is already evident. The prevalence of delusions ranges between 10% and 20% [16,78].

Cognitive decline is part of the PDD/DLB complex; an annual incidence of about 10% is reported [16].

While these three elements constitute the most well-known standards in the field of neuropsychiatry in PD, other disturbances are equally crucial due to their impact on therapy.

### 4.3. Somatic Symptoms and Functional Neurological Disorder

A pervasive mental disorder in PD has been described in detail in many studies by expert centers: the presence of somatic symptoms and related disorders [79,80,81,82,83,84,85,86,87,88]. These encompass manifestations historically referred to as hypochondria, Briquet’s syndrome, hysteria, conversion disorder, somatoform disorders, psychosomatic disorders, and psychogenic disorders [20,54,89]. These disturbances almost typically precede the onset of motor symptoms by several years, primarily accompanying the early stages of the disease in predisposed patients. They are less evident, but not always absent, in the more advanced stages associated with cognitive decline. Often, these disturbances involve abdominal discomfort; sensations of bloating; dyspepsia; various types of pain that disappear when patients are distracted; pill dysphagia without difficulty in swallowing food and liquids; physiologically and pharmacologically impossible intolerances to administered medications; and impossible complications, such as dyskinesias occurring concurrently with clearly OFF periods when motor therapy has already been initiated [20,85,86].

Years before motor symptoms emerge, patients may present hypochondriacal manifestations or conversion disorders. In some cases, motor manifestations of PD, such as tremors or distractible and inexplicable paresis; a long history of conditions categorized as fibromyalgia [90] or chronic fatigue syndrome; and painful syndromes devoid of pathological substrates, such as Reflex Sympathetic Dystrophy, episodes of *Globus pharyngis* (formerly known as hysterical bolus), and pseudoseizures (psychogenic epileptic seizures), all fall under the category of Medically Unexplained Syndromes (MUS) [20,54]. Some authors estimate the prevalence of these disorders to be 10–20% [20], while others suggest that it may exceed 30–59% [85,87,91]. Figure 1 compares the timing of cognitive, psychiatric disorders and FND-SSD in two different clinical scenarios in PD.

The presentation and progression of FND in PD patients can differ according to the degree of cognitive impairment. In the first scenario, symptoms begin in a prodromal/early phase, decrease after the onset of PD motor signs, and reappear in subsequent stages, mixed with somatic delusions and other psychotic disturbances. In a more benign scenario, FND gradually fades after the onset of cognitive decline and other psychotic symptoms.

However, in the psychopathological evaluation, a significant lack of critical insight regarding the disorders emerges, sometimes with psychotic elements, to the extent that it has been hypothesized that somatic symptom disorders in somatic patients are akin to true somatic hallucinations [46,50,54]. In some cases, due to their presentation, forms of somatosensory hallucinations (referred to as tactile or haptic hallucinations) are recognized [92]. Rarely, in cases of PD where cognitive decline has already occurred, a somatic delusion can manifest, involving hallucinatory sensations of deformity, delusions of parasitic infestation (Ekbom’s Syndrome), delusions of multiple allergies, or delusions of having necrotic body parts (Cotard’s Syndrome) [16,93,94,95,96,97].

### 4.4. Impulse Control Disorders (ICDs) and Their Recent Correlation with Bipolar Disorders

ICDs encompass the emergence of compulsive gambling; hypersexuality, including paraphilic manifestations; increased compulsive shopping; and bulimia, but also other compulsive or coercive behaviors, such as aggressive driving and obsession with do-it-yourself projects, exercise, household cleaning, or gardening [98,99,100,101]. The prevalence of ICDs ranges from 8% to 50% [21,100,101,102,103].

A similar phenomenon, albeit not related to other ICDs, is Dopa Dysregulation Syndrome (DDS) [104], which leads to an uncontrollable (and unjustified given the disease stage) desire to take L-dopa tablets continuously.

These manifestations have been observed in conjunction with the introduction of newer dopamine agonists (DA), such as ropinirole, pramipexole, and rotigotine. However, they were already described for the older ergoline dopamine agonists (bromocriptine, cabergoline, pergolide) [105,106,107], and with the introduction of L-dopa [104,108]. DA triggers ICDs, and their incidence increases from 5–7% to 18%, up to 50%, according to some studies [21,101,109]. In PD caused by PRKN and GBA mutations, these disorders are very common and appear before motor symptoms and, of course, before exposure to DA or L-dopa therapy [23,110]. However, the presence of ICDs in patients not exposed to DA or L-dopa raises the issue of predisposition and what dysfunction in brain circuits leads to the appearance of ICD manifestations.

It is essential to emphasize that these same behaviors are observed in patients with bipolar spectrum disorder (BSD) and constitute the central diagnostic core [70].

### 4.5. Preexisting Bipolar Disorder and Impact on PD

In the last four years, numerous studies conducted on large patient populations have shown that BSD precedes the onset of PD, with odds ratios (OR) ranging from 3 to 5 [9,111,112,113,114,115,116,117,118,119,120,121]. A single genome-wide association study [122] challenged these conclusions shortly after the publication of one of the last epidemiological studies [116]. Still, ensuing studies showed instead a correlation [60,123]. Anecdotal literature already existed before these systematic studies [124,125,126], and earlier epidemiological studies [112,116,119] and studies in Scandinavian countries had already shown similar findings [127,128]. As early as 2006, it was reported that the presence of BSD poses a significant risk of failure for PD surgical therapies, providing a clear indication that a portion of the PD population may present this predictive phenotype [129,130,131]. Recent epidemiological studies have definitively refuted the hypothesis that the coexistence of PD and BSD is merely a coincidence [111,112,116,117,118,119,132].

Therefore, BSD preceding PD constitutes a specific clinical phenotype: in these patients, the incidence of ICD increases, often appearing before motor symptoms, and the incidence of dopamine agonist withdrawal syndrome (DAWS) increases, as does the incidence of delirium and SSD [9,117]. The incidences of cognitive decline and hallucinations, on the other hand, remain unchanged. When considering BSD, it must be remembered that the current classification envisions only three main variants of the disorder—Bipolar I, Bipolar II, and Cyclothymia [70]. Hypomania often surfaces in studies on PD [9,16,133], and a more complex classification of BSD was provided by one of the foremost experts of BSD, Akiskal [134]: his classification proposes six variant aspects of BSD, with a specific group characterized by cognitive disorders (Table 2 reproduces his classification). Despite the fact that this classification is not accepted by ICD-11 and DSM-5-TR [135], the recent literature is convincingly evidencing its consistency [117,121]. The occurrence of neurodegenerative disorders in BSD is supported by powerful epidemiological studies [22,111,113,115,116,118]. The type VI described by Akiskal consists of a specific phenotype, termed Fronto Temporal Phenocopy Syndrome, or the proper phenotypes of neurodegenerative disorders. Recent reports, however, seem to dismiss the existence of Phenocopy Syndrome, suggesting that neuropathological evidence does not show relevant differences from the proper phenotypes of neurodegenerative disorders.

The coexistence of PD and BSD suggests that dysfunction in neural circuits is likely common in both pathologies and suggests new hypotheses for research into mechanisms and therapies [9,115,116,117,118]. However, it also raises the issue of how much damage media campaigns have caused, until a few years ago, to a dimension that required prior epidemiological and scientific clarifications. This has led to a rush of compensation claims against doctors and pharmaceutical companies that were not supported by proven demonstrations of preexisting conditions. In conclusion, BSD is a significant predictive factor for the onset of PD, challenging earlier media oversimplifications and emphasizing the need for more nuanced discussions in both the clinical and public domains [136,137,138,139].

## 5. Cognitive Fluctuations

In the early 1990s, a new reality emerged: a type of dementia was identified that included PD motor symptoms and was characterized by the same Lewy bodies (aggregates of alpha-synuclein) found in PD. This type of dementia accounts for 10–25% of all dementia and is referred to as dementia with Lewy bodies (DLB) [140]. It is characterized by cognitive decline, with rare involvement of the mnemonic domain, and instead frequent involvement of visuospatial and executive functions, with early onset of hallucinations, delusions, hypersensitivity to neuroleptics (which can also be lethal in these patients [141]), parkinsonian motor symptoms, electroencephalographic (EEG) alterations, orthostatic hypotension, and rapid eye movement sleep behavior disorder (RBD), as well as cognitive fluctuations (CF). CF, along with recurrent visual hallucinations, motor symptoms, RBD, and evidence of cognitive decline (not so severe as to be defined as dementia) constitute the core criteria according to international guidelines for DLB diagnosis [140]. CF manifest as brief or prolonged episodes of disconnection from the external world, ranging from apparent absences akin to “trances” to confused, dreamlike, and hallucinatory episodes [142]. CF are assessed with scales devised and validated for interrater reliability, like the Clinician Assessment of Fluctuations [143], which also evaluates dreamlike and oniric hallucinations. Still, recent studies have suggested that bedside tests of fluctuations of working memory could support diagnosis.

CF are linked to EEG abnormalities, consisting of an inscription of slow theta activity on the background alpha EEG activity of the patient. This theta activity is also defined as cortico-thalamic dysrhythmia [46]. These fluctuations are often misinterpreted as transient ischemic episodes or epileptic seizures by less experienced physicians [144,145].

Since the same disturbances appear later in the progression of the disease in typical PD, it was decided that DLB should be classified as a condition where cognitive and psychiatric symptoms appear first, together with—or, at most, one year after—the onset of motor symptoms. Meanwhile, in PD with dementia (PDD), motor symptoms occur at least one year before cognitive and psychiatric symptoms. Nevertheless, due to the similarity in alpha-synuclein aggregates and the type of neurodegeneration, DLB and PDD are commonly considered as a continuum of the same disease [146]. Figure 2 compares the timing of cognitive and psychiatric disorders in PD and DLB.

The figure emphasizes the prominent and early psychiatric and cognitive burden of DLB patients compared to PD patients. However, PD patients can present persistent, although more “subtle”, psychiatric symptoms (in particular, FND) since the earliest phase of the disease, often before the onset of motor symptoms.

## 6. Therapeutic Implications of the Presence of Psychiatric Abnormalities

The management of psychiatric complications in PD deserves attention and clinical competence because it has a central impact on therapy management [147]. Another important aspect of therapeutic management is the overlap of psychosomatic disorders. This overlap can complicate the acceptance of chronic therapy, leading to unconscious resistance and the emergence of treatment-related disorders. The molecular mechanism of the drug should not explain this phenomenon or be entirely inconsistent with the drug’s duration (half-life, AUC). In our clinical practice, we often encounter common psychosomatic disturbances like abdominal discomfort with a sensation of bloating or constipation attributed to L-dopa (as illustrated previously). While L-dopa does cause modest nausea in less than 10% of patients upon initial administration, this side effect disappears after a few days. This is because dopamine is the neurotransmitter of the brainstem chemoreceptor area and does not show any toxicity to the digestive system [148]. On the other hand, constipation is directly related to PD itself rather than a consequence of therapy. Another refusal phenomenon manifests as “L-dopa phobia” [149], which leads many patients to reject L-dopa, despite all the recent literature (from the last 20 years) showing that early introduction of L-dopa (with great care in dosing) produces better results in disease progression and the onset of fluctuations and dyskinesias [150].

However, the problems to be managed are often much more complex, depending on the poor capacity for self-awareness (introspection, insight) that prevents patients from recognizing the effect of therapies on their motor abilities, muscle tone disturbances (dystonia), or even tremors.

### 6.1. Mood Disorders

Sometimes, manifestations of anxiety that include panic attacks become unmanageable for family members and generate unjustified increases in drug doses, leading to the early appearance of involuntary movements. It is true that “psychological OFF” phenomena are recognized. They occur when the effect of L-dopa wears off, and anxiety or agitation states reappear. However, these “psychological OFF” states are easily recognizable because they always have a close temporal relationship with L-dopa administration [151,152,153]. If anxiety is considered an OFF non-motor symptom, it should be treated with levodopa or dopamine agonists [14,25]. The appearance of anxiety during a ”OFF-period” should alert the clinician to DDS, since this can precede up to 35% of DDS cases [28]. However, anxiety could be independent of the dopaminergic treatment and, in that case, treatment with selective serotonin reuptake inhibitors (SSRI) and serotonin and norepinephrine reuptake inhibitors (SNRI) is indicated [14,25]. A similar approach is considered for depression treatment. The Movement Disorder Society proposes dopamine agonist drugs to treat depression in PD, and in particular, the last evidence-based medicine update considers pramipexole as clinically useful [154,155]. Although antidopaminergic medication should always be considered as the first-line treatment for depression, classic antidepressive agents could be accounted for [154,155]. Among antidepressants, no class demonstrates significative effectiveness compared to others [154,155]. Venlafaxine is considered clinically efficacious and benefits from combined serotonin and norepinephrine actions, compared to other SSRIs or SNRIs [154,155]. SSRI and SNRI rarely cause motor symptoms to worsen but are known to cause worse tremors [154,155]. Despite similar results, tricyclic antidepressants are limited due to their anticholinergic effect and the related adverse effects on cognitive functions and balance [154,155]. The choice is strictly related to therapy customization and varies from patient to patient, mostly depending on the features of the depression. The challenge for the clinician is the first-choice treatment. It might be useful to treat depression that is recognized as an off-motor symptom with antiparkinsonian drugs, whereas SSRI or SNRI may be useful to treat a depressive personality. Finally, other authors have documented a huge benefit for mood provided by non-pharmacological therapy, mostly including physical activity [156,157].

### 6.2. Psychiatric Complex

Visual hallucinations can be distinguished as minor (“extracampine, i.e., those occurring outside the visual field”) or major (complex) [47,158]. Moreover, it is important to define whether hallucinations are related to dopaminergic therapy or to disease progression. Hence, non-stressful extracampine hallucinations may not require pharmacological management, whereas complex hallucinations should require the titration of dopamine agonist or levodopa dosage, then atypical antipsychotics should be evaluated, mostly in concurrence with neurocognitive disorders [16,158,159]. The challenge for the clinician is to understand how much the current pharmacological treatment impacts the symptomatology. Often, PD patients show fragility in terms of neurodegeneration due to multi-pharmacological therapy, including SSRI, SNRI, and anti-cholinergic drugs, as these are known to trigger hallucinations [159]

As noted above, psychosis and ICDs could severely impact patients’ and caregivers’ QoL. Both symptoms are significantly influenced by the history of the disease, the pharmacological treatment, and some genetic forms. To begin with, current guidelines recommend at first decreasing the dopaminergic therapy to treat psychosis and ICDs, which are significantly induced by dopamine agonists and polytherapy [158,160,161]. Then, both disorders should be considered for therapy individualization. ICDs have historically been considerably ascribed to high dosages of dopamine agonists, even in young patients, thus causing a severe impact on daily life [162]. That should account for the best medical pharmacotherapy, which involves considering the switch from dopamine agonists to other antiparkinsonian agents, adding non-pharmacological treatments such as cognitive behavioral therapy, adding atypical antipsychotics if needed, and finally, considering advanced therapy [160]. A different approach is advisable for older PD patients affected by psychosis and dementia. These kinds of patients should be managed through a review of dopaminergic drugs, avoiding dopamine agonists and Catechol-O-methyltransferase (COMT) inhibitors, dealing with cognitive impairment, and using more easily atypical antipsychotics [155,158,161].

Antipsychotics should be used with caution in parkinsonisms, mostly in DLB but even in PD. Currently, quetiapine and clozapine do not affect motor symptoms and are considered safe in PD, albeit with some recommendations [154,158]. Mainly, quetiapine administration requires routinary EKG-QT trait monitoring, while clozapine is reported to cause agranulocytosis, limiting it from being used in complex pharmacological and legal management [163,164,165]. Typical antipsychotics and lithium are reported to worsen motor symptoms in parkinsonisms and some authors debate their relationship to iPD in BSD patients. However, drug-induced parkinsonism shows preserved dopaminergic functional imaging compared to idiopathic parkinsonism, and different studies have demonstrated the strong relationship between BSD and idiopathic PD [9,110,115].

In PD, typical and atypical antipsychotics with partial agonism for D2 receptors, such as olanzapine, risperidone, asenapine, aripiprazole, brexpiprazole, and ziprasidone, all result in severe worsening of PD [165,166] and possible neuroleptic malignant syndrome, which may end in a fatal outcome [140].

Pimavanserin is also used in the USA, the UK, and Japan. However, this drug did not pass evaluations by the European Medicines Agency (EMA) [167,168], as the statistical evaluations in the European assessment’s double-blind trial did not reach sufficient evidence. Therefore, the pharmacological management of PD psychosis is still off limits to quetiapine and clozapine. Moreover, the use of clozapine seems unevenly distributed among European countries, as it is used more often in France and Italy than in other countries [169,170].

Even more complex is the problem of the response to advanced therapies: the presence of psychiatric disorders constitutes, in itself, a relative contraindication to DBS, and the literature indicates that patients with hypomanic or bipolar spectrum disorders poorly benefit from DBS [9,128,130]. However, the subthalamic nucleus-DBS (STN-DBS) considerably lowers the L-dopa equivalent dose (LED); therefore, it could reduce drug-related psychosis and ICDs [171]. Thus, the patient’s psychiatric profile could significantly influence the choice of advanced therapy.

A pharmacoepidemiologic study in different countries is needed, as well as the implementation of further antipsychotic drugs devoid of dopamine receptor interactions. Unfortunately, since the early evidence of the efficacy and safety of clozapine was found, the search for new drugs to be safely used for the treatment of psychosis in parkinsonism has been unfruitful, except for pimavanserin [172], which the EMA should reevaluate.

In addition to its importance for the future development of innovative therapeutic approaches, the presence of psychiatric disorders in PD assumes central importance in current therapeutic organizations. A significant percentage of PD patients experience dysfunction in recognizing consensual reality, awareness of their bodily dimensions, and emotional influx control. However, therapy cannot be limited to pharmacological prescriptions, and physical therapies cannot be separated from a psychotherapeutic approach.

## 7. Conclusions and Considerations About Psychotherapy and Physiotherapy

The integration of physiotherapy with elements of psychotherapy is essential, especially in the context of SSD-FND [173]. Still, this integration finds its most epidemiologically significant outlet in PD physiotherapy [174,175].

Activities that engage family members and caregivers are crucial as they help alleviate psychological stress and foster a supportive environment [176,177,178]. Support groups like patient associations are also important. The benefits observed from group activities—such as dance, group gymnastics, music therapy, and theatrical acting—can be attributed to their positive impact on the psychodynamic dimension and the development of adult defenses, which help to preserve identity rather than allowing regression to infantile defenses, like FND-SSD. Neurophysiological and functional neuroimaging studies have demonstrated how artistic involvement can produce significant changes in the functioning of the central nervous system through synaptic plasticity mechanisms [179,180]. Among the brain areas that activate in response to a piece of music, the reward or pleasure circuits play a crucial role, as their activation leads to a state of well-being [181,182].

Physiotherapeutic strategies should be planned in relation to guidelines and the individual patient’s profile to ensure adherence and the achievement of results [183]. It is essential to explain to patients how physical symptoms are interpreted. The goal is to build an alliance between physiotherapists and patients to improve the quality of life of those with PD.

Furthermore, predisposition to somatizations leads to maladaptive behaviors in relation to the disease, such as the development of anxiety, hypochondria, thanatophobia, fear of the illness, or, conversely, denial of the disease [54,86,184].

For patients with SSD and FND, but not those suffering from PD, physiotherapy should monitor motor response. Psychotherapy and pharmacotherapy should be synergistically directed toward resolving concurrent neuropsychiatric disorders. While the relatively slow effect limits the psychodynamic approach (psychoanalysis), short-term psychotherapeutic techniques used in the treatment of psychosomatic disorders may also find application in patients with PD, including short term psychodynamic psychotherapy (STPP), cognitive behavioral therapy (CBT), and treatment of anxiety and physical symptoms (TAPS) [185,186].

The demonstration of the effectiveness of these approaches, such as mindfulness techniques, hypnosis, stress management, and group psychoeducation, is currently occasional or anecdotal [187]. However, considering the knowledge acquired about psychiatric disorders in PD, they represent an essential future direction for the organization of outpatient services, care, and compensation for PD, which is far more relevant than therapeutic pathways directed solely at pharmaceutical distribution.

Finally, a recent meta-analysis explored the effectiveness of non-pharmacological treatments for the management of disorders associated with PD [188]. Among the treatments included in the study were brain electrical activity stimulation therapies (e.g., transcranial magnetic stimulation, transcranial direct current stimulation), various standardized or semi-standardized physical practices (aerobics, balance exercises, dance), various types of psychotherapy alone or in combination with occupational therapy, music therapy, etc. The most effective treatment judged in the study was dance therapy, followed by the Lee Silverman Voice Treatment-BIG physiotherapy technique, and in third place, CBT.

## Figures and Tables

**Figure 1 brainsci-14-01233-f001:**
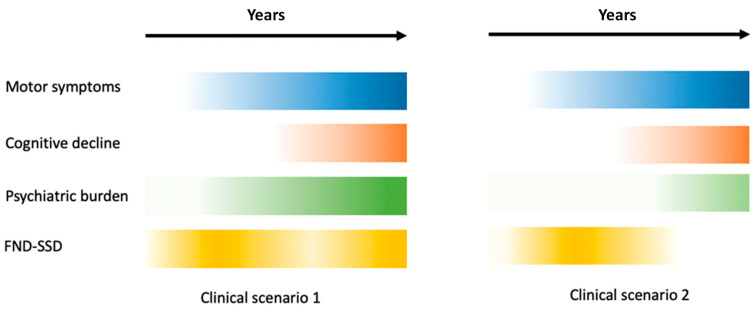
The clinical course of functional neurological disorders and other non-motor and motor symptoms in patients with typical PD in two different clinical scenarios. The intensity of the color represents the severity of the symptoms: the more intense the color, the more severe the symptoms. Abbreviations: FND-SSD: functional neurological disorders–somatic symptom disorder.

**Figure 2 brainsci-14-01233-f002:**
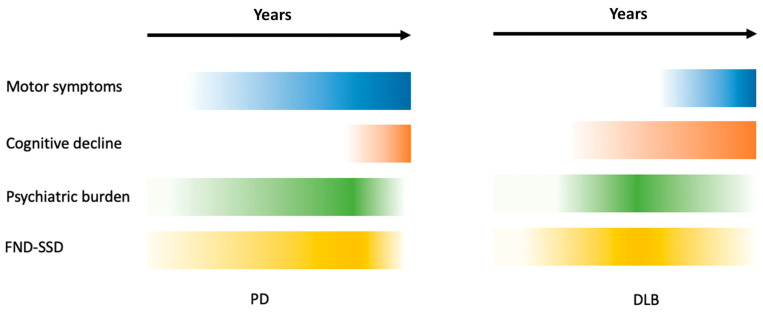
The progression of clinical symptoms in PD and DLB. The panel illustrates a schematic representation of the temporal course of psychosis, functional disorders, motor symptoms, and cognitive impairment in PD and DLB. The intensity of the color represents the severity of the symptoms: the more intense the color, the more severe the symptoms are. The time, in years, depends on the rate of progression of the disease, which will be different between Typical PD and DLB not linked to genetic mutations, and will be shorter in GBA genetic mutations, LRRK2, and Atypical Parkinsonism. Abbreviations: Parkinson’s Disease; DLB: dementia with Lewy bodies; FND-SSD, functional neurological disorders–somatic symptom disorder.

**Table 1 brainsci-14-01233-t001:** Neuropsychiatric and cognitive disorders in PD.

**Preceding motor symptoms**	
Bipolar spectrum disorders (BSD)	Present 4-to-6-fold higher risk of developing PD
Somatic symptom disorders (SSD)	Precede or accompany motor symptoms in 10–59% of patients
Impulse control disorders (ICDs)	Precede motor symptoms in s genetic forms, are associated with BSD
Depression	Precede PD, described in XX century literature
**Accompanying the onset of motor symptoms**	
Simple hallucinations	
Depression	
**Triggered by medical treatment**	
Simple and complex hallucinations	In the elderly
Impulse control disorders (ICDs)	Linked to monogenic variants and BSD
Structured delusions	Rare, associated with BSD, genetic variants, dementia
**Emerging with the progression of the disease**	
Hallucinations	Strictly associated with neuropathology
Cognitive fluctuations	Associated with cognitive impairment, precede full evidence of dementia
Dementia	Depending upon age, genetic mutations

**Table 2 brainsci-14-01233-t002:** Division of bipolar disorder according to Akiskal’s classification.

Akiskal Classification	Clinical Characteristics
Bipolar I	Full-blown mania
Bipolar I 1⁄2	Depression with protracted hypomania
Bipolar II	Depression with hypomanic episodes
Bipolar II 1⁄2	Depression associated with cyclothymic temperament
Bipolar III	Hypomania due to antidepressant drugs
Bipolar III 1⁄2	Hypomania and/or depression associated with substance use
Bipolar IV	Depression associated with hyperthymic temperament
Bipolar V	Recurrent depression that is admixed with dysphoric hypomania
Bipolar VI	Late-onset depression with mixed mood features, progressing to dementia-like syndrome

## Data Availability

Not applicable.

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
