# Peer review of "Psychiatric Disorders and Cognitive Fluctuations in Parkinson’s Disease: Changing Approaches in the First Decades of the 21st Century"

_brainsci, 2024, doi:10.3390/brainsci14121233_

Round 1
Reviewer 1 Report
Comments and Suggestions for Authors
1. This article explores an important topic and does so from a variety of perspectives. However, the structure lacks cohesion, with sections that sometimes appear disjointed. I recommend reorganizing sections to address early- and late-stage symptoms, ensuring smooth transitions between discussions of symptoms and disease progression. More descriptive headings would also help guide readers through each section’s main points. The following article provides references for the author: Khan et al., (2017). A comprehensive overview of the neuropsychiatry of Parkinson's disease: A review. Bulletin of the Menninger Clinic, 81(1), 53-105.; Yu & Wu "Mild cognitive impairment in patients with Parkinson’s disease: An updated mini-review and future outlook." Frontiers in aging neuroscience 14 (2022): 943438.
2. The discussion section lacks robust data to support some key conclusions. Adding quantitative data or empirical evidence directly supporting the primary assertions would enhance credibility. Consider using tables or figures to clearly illustrate the prevalence of various symptoms and their progression over the disease course. This would also increase the manuscript's overall persuasiveness.
3. In discussing treatment approaches, consider providing more specific insights into the practical application and challenges of these therapies in clinical settings. Highlighting the indications and contraindications for different treatments would be valuable for clinical readers, adding practical depth to the paper's contribution to treatment decision-making in PD.
4. Does "Cognitive Fluctuations" in the title of this study refer to "Cognitive Dysfunction"? If so, it would be advisable to change "Fluctuations" to "Dysfunction." The term "Fluctuations" typically implies short-term variations, often seen as a specific feature in certain conditions, and may not be appropriate here.
5. One detail I find puzzling is whether the "p" in "Typical Parkinsonism" or "Atypical Parkinsonism" should be uppercase or lowercase.
Overall, this study offers a thorough review of the psychiatric and cognitive symptoms in PD, detailing psychopathological changes across disease stages and related treatment challenges. However, some sections are lengthy, and certain terms, like "Cognitive Fluctuations," need clearer definitions to avoid ambiguity. Streamlining the historical background and clarifying the methodology could enhance readability. With these refinements, this paper could be a valuable contribution to understanding and managing PD's non-motor symptoms. It is recommended that the author be allowed to make revisions, after which the manuscript can be re-evaluated for potential acceptance.
Comments on the Quality of English Languageok
Reviewer 2 Report
Comments and Suggestions for Authors
This is an interesting review , but can be improved in some aspects:
1.the type of review should be indicated , as narrative.
2.in the section "Pathophysiological mechamisms..." it would be clearer and approriate to indicate a new section by a new heading about DLB/PDD
3.again a new appropriate heading would be good before the now beginning text on "PD,PD disease..." , in fact this section focusing on somatic symptom disorder within these various subclassifications
4. in figure 1 it remains unclear which subtype of PD ( typical PD??) is meant, please clarify in text to figure and figure. Similar üroblem in figure 2, which is in principle rather helpful.
5. for text and table 2 it should be more clearly mentioned that the Akisklapreoposal did not make it into esatblished classifications ICD 11 and DSM V.6
6.the "media oversimplification.." should be more clearly expalined. what was the oversimplification and potentially which media
7.The heading "Impact on Physiotherapy" appears not fully matching the text of this section, better seemed to me "Considerations about psychotherapy and physiotherapy".
In this section 2 sentences or parts of are hard to understand: a: "..aggression that the disease exerts.." b: "..production of ...."
Reviewer 3 Report
Comments and Suggestions for Authors
The authors emphasize the need for a holistic understanding of Parkinson's Disease, wherein cognitive and psychiatric dimensions are valued alongside motor symptoms. An extremely relevant topic has been touched upon. The authors did an extensive job summarizing recent work in this field. The manuscript is well written, the introduction provides sufficient background. However, some questions require clarification.
1. I recommend that you enter the numbering of sections and subsections.
2. Line 162: “Dementia with Lewy Bodies”. The abbreviation has already been introduced (DLB, line 146).
3. Fig. 1, Fig. 2. The diagrams look incomplete. Maybe you should indicate the age periods on the arrow “years”?
4. Fig. 1. Please add a transcript of the abbreviations.
5. Line 184: “PD, PD disease; DLB, Dementia with Lewy Bodies; FND-SSD, Functional Neurological Disorders- Somatic Symptom Disorder”. Is this the name of the section?
6. Line 211 – 215. Why did the authors separate these sentences into paragraphs?
7. It seems necessary to add a section “Conclusions”.
8. Please arrange the references in accordance with the requirements of the journal.
